# Influence of Failure-Load Prediction in Composite Single-Lap Joints with Brittle and Ductile Adhesives Using Different Progressive-Damage Techniques

**DOI:** 10.3390/polym16070964

**Published:** 2024-04-02

**Authors:** Yung-Cheng Chuang, Cong-Sheng Su, Yu-Jui Liang

**Affiliations:** Department of Aeronautics and Astronautics, National Cheng Kung University, Tainan 701, Taiwan; p46111397@gs.ncku.edu.tw (Y.-C.C.); p46111193@gs.ncku.edu.tw (C.-S.S.)

**Keywords:** single lap joint, cohesive zone model, virtual crack closure technique, Abaqus

## Abstract

The usage of adhesively bonded joints, such as single-lap and double-lap joints, is increasing rapidly in aerospace composite structures as a popular alternative to bolts and rivets. Compared to the conventional joining methods such as fastening and riveting, adhesive-bonding technology better prevents damage to composite structures due to the smooth configuration and the mitigation of stress concentration around holes. In this work, the built-in progressive-damage-modeling techniques in Abaqus, including the cohesive zone model (CZM) and the virtual crack closure technique (VCCT), are used to predict the strength and progressive failure of composite single-lap joints subjected to tensile loading. Modeling of an adhesive layer by using a zero/non-zero-thickness cohesive element, cohesive surface, and VCCT is investigated, as is the effect of brittle and ductile adhesives. Two-dimensional finite-element models with different damage-modeling strategies are performed in this study. The failure-load predictions are compared with the experimental results obtained from the literature. For the ductile adhesive, the predicted failure loads using a zero/non-zero-thickness cohesive elements and a cohesive surface are all shown to be in good agreement with the experiments. However, the VCCT technique predicts higher failure loads. For a brittle adhesive, on the other hand, the predictions by zero/non-zero-thickness cohesive elements and cohesive surfaces reveal notable deviations compared to the experimental results. In contrast to the ductile adhesive, the VCCT technique is revealed to be accurate in predicting the brittle adhesive.

## 1. Introduction

Advanced composite materials are increasingly replacing traditional metallic materials in various applications due to their high strength, stiffness-to-density ratios, and design flexibility, as well as their fatigue and corrosion resistance. These outstanding material properties often make them the preferred option in various sectors such as the aerospace, space, automobile, civil, and marine industries. In recent years, the usage of adhesive bonding, which is one of the joining methods in the structural components of composites, has increased rapidly in engineering applications due to many advantages such as light weight, lower cost, and improved damage tolerance over conventional mechanical-joining methods. Since the increasing demand of adhesive bonding, particularly in aerospace composites, in recent years, many researchers have been investigating the performance of adhesively bonded composite joints, resulting in the development of material, geometric modification of adherend and adhesive joining techniques, and novel analyzed methods. The comprehensive background and progress of the adhesively bonded joints in composite materials over the past few decades are provided in the review papers [1,2].

To further understand the parameters that affect the performance of the adhesively bonded composite joints, analytical methods and finite-element methods (FEM) are often used in the analysis. Analytical methods predict the performance of the joints easily, quickly, and accurately when geometries of joints are simple and idealized, but they might be inappropriate for some complex structures [3,4,5,6,7,8,9]. However, due to the increasing demand for analyzing large and complex composite bonded joints, it is difficult to obtain an overall system of governing equations in the mathematical formulation. Moreover, the experimental tests are often time-consuming and expensive. Therefore, the FEM is an appropriate method because of the capability to analyze complex geometries, material nonlinearity, and different boundary conditions. One thing needs to be noted—that a limited number of degrees of freedom in the finite-element simulation must be considered to reduce the execution time by a reasonable amount. Several finite-element analyses of adhesive joints can be found in the literature [10,11,12,13,14,15,16,17,18].

In recent years, the damage-mechanics approach has been attracting attention in the usage of damage modeling for composite bonded joints, as it performs the most accurate failure predictions. Significant progress has been achieved in developing progressive-damage analysis (PDA) methodologies, which are able to capture damage initiation and evolution in composites and predict their strength and durability for various lay-ups and geometries [19,20,21,22,23,24,25,26]. The most commonly used progressive-damage model is called the cohesive zone model (CZM); the concept was first proposed by Barenblatt [27] and Dugdale [28], who described the damage under static loads with no need of an initial crack. Since then, the CZM was improved and has been widely used in the application of crack propagation to almost any kind of material, from micro to macro scales [29,30,31,32]. An important feature of CZM is that it is easy to implement in conventional finite-element software to model progressive-failure behavior in various materials, including composite bonded joints [10,14,16]. The standard and extended FEM were adopted in strength predictions of single- and double-lap joints in reference [33]. A recent research paper proposed a bio-inspired single-lap joint with CFRP and metal using the CZM method [34], where the same CZM method was also applied to investigate the effect of adherend notching on strength performance [35]. On the other hand, another approach is called the virtual crack closure technique (VCCT), which assumes that the energy required for new crack-face creation is equal to that required to close the crack back to its original length [36,37]. The VCCT is also widely used to simulate the crack propagation and predict the failure load of composite bonded joints such as the single-lap joint (SLJ) [38,39,40]. Although VCCT has some limitations regarding the existing initial crack and remeshing issue, it often has better computational efficiency compared to the CZM.

As mentioned above, CZM and VCCT have been utilized for strength prediction and bonding performances in various types of composite bonded joints. However, the effect of using different PDA techniques on composite-joint-bonding performances under different adhesive types is still not fully understood and has only been proposed in limited research. Commercial structural adhesives from strong yet brittle to less strong yet ductile are widely used in structural adhesive boding, depending on different applications, especially in aerospace structures. The adhesive properties play a significant role in determining joint strength. However, a stronger adhesive doesn’t always result in a higher joint strength. For instance, a highly strong yet brittle adhesive may concentrate stresses excessively at overlap corners without enabling stress redistribution to less stressed areas. Consequently, the average shear stress at failure remains remarkably low. On the other hand, adhesives characterized by high ductility and low modulus typically exhibit lower strength. Nevertheless, they tend to distribute stresses more evenly along the bondline and undergo plastic deformation, thereby rendering the joints stronger compared to those formed with strong yet brittle adhesives. To push forward the engineering application of adhesively bonded joints, it is meaningful to investigate the influence of failure-load prediction in composite bonded joints with different progressive-damage techniques, considering, in particular, two different brittle and ductile adhesives. Therefore, the main objective of the present effort is to investigate the influence of using different modeling techniques (CZM and VCCT) on failure-load prediction of composite SLJ with different brittle and ductile adhesives. In this article, the CZM and VCCT methodologies are introduced first, followed by a description of the 2D SLJ finite-element model. The built-in capabilities of CZM in Abaqus, including cohesive contact and cohesive element, are considered for the strength prediction of the SLJ model, as well as for the VCCT. Last, the predicted results obtained from cohesive contact, cohesive element, and VCCT are compared with the experimental data from the literature [35] and followed by the conclusions of this study.

## 2. Methodology

### 2.1. Cohesive Zone Model (CZM)

As mentioned above, the CZM, which has been widely used in damage modeling, is considered to be the most popular method for simulating the progressive damage of composite adhesively bonded joints. The proliferation of the CZM methodology is greatly aided by the ease of its implementation in a conventional finite-element framework such as Abaqus 2023 [41]. The main merit of the CZM lies in the elimination of stress singularity at the crack tip, by applying the softening relation between stress and relative displacement of crack surfaces to simulate the elastic behavior up to $t_{n}^{0}$ in tension or $t_{s}^{0}$ in shear and the subsequent softening up to failure, as shown in Figure 1. The bilinear CZM, as shown in Figure 1, is used to simulate the damage initiation and propagation of the adhesive layer in this work. It assumes an initial linear behavior followed by linear degradation. Damage initiation can be specified by different criterion, either stress- or strain-based. The quadratic nominal stress criterion is used herein, which is expressed as follows:(1)$${\left(\frac{\langle t_{n}\rangle }{t_{n}^{0}}\right)}^{2}+{\left(\frac{t_{s}}{t_{s}^{0}}\right)}^{2}+{\left(\frac{t_{t}}{t_{t}^{0}}\right)}^{2}=1$$
where the $t_{n}$, $t_{s}$, and $t_{t}$ are the tension, shear, and tearing stresses, respectively. The $t_{n}^{0}$, $t_{s}^{0}$, and $t_{t}^{0}$ are the maximum values of nominal stress in tension, shear, and tearing directions, respectively. The expression of $\langle t_{n}\rangle$ is:(2)$$\langle t_{n}\rangle =\left\{\begin{array}{c} t_{n},\;t_{n}>0\\ 0,\;t_{n}<0 \end{array}\right.$$

Note that a normal compressive stress does not initiate damage. Once the failure criterion is reached, the material stiffness initiates a degradation process while a damage variable *D* (0 < *D* < 1) is introduced. The material constitutive relationship becomes:(3)$$\left\{\begin{array}{c} t_{n}\\ t_{s}\\ t_{t} \end{array}\right\}=\left[\begin{array}{ccc} (1-D)K_{nn} & 0 & 0\\ 0 & (1-D)K_{ss} & 0\\ 0 & 0 & (1-D)K_{tt} \end{array}\right]\left\{\begin{array}{c} \delta_{n}\\ \delta_{s}\\ \delta_{t} \end{array}\right\}$$
where the $K_{nn}=E/t_{A}$, $K_{ss}=K_{tt}=G/t_{A}$ and $K_{ns}=0$ for the thin adhesive layer. The $t_{A}$ is the thickness of the adhesive layer. The $\delta_{n}$, $\delta_{s}$, and $\delta_{t}$ are the tension, shear, and tearing displacements, respectively. The linear-softening-damage variable *D* is defined as follows:(4)$$D=\frac{\delta_{m}^{f}\left(\delta_{m}-\delta_{m}^{0}\right)}{\delta_{m}\left(\delta_{m}^{f}-\delta_{m}^{0}\right)}$$
where the $\delta_{m}^{0}$ represents the effective displacement of damage initiation, $\delta_{m}^{f}$ represents the effective displacement of final failure, and $\delta_{m}$ represents the current effective displacement, which is computed as follows:(5)$$\delta_{m}=\sqrt{{\langle \delta_{n}\rangle }^{2}+{\delta_{s}}^{2}+{\delta_{t}}^{2}}$$

The power-law criterion is used in this work under mixed-mode loading conditions. The expression is given as follows:(6)$${\left\{\frac{G_{I}}{G_{IC}}\right\}}^{\alpha }+{\left\{\frac{G_{II}}{G_{IIC}}\right\}}^{\alpha }+{\left\{\frac{G_{III}}{G_{IIIC}}\right\}}^{\alpha }=1$$
where the α is the power-law parameter, and $G_{I}$, $G_{II}$, and $G_{III}$ are the mode I, II, and III strain energy release rate (SERR), respectively, while $G_{IC}$, $G_{IIC}$, and $G_{IIIC}$ are their corresponding critical values where they represent the areas under the bilinear CZM.

### 2.2. Virtual Crack Closure Technique (VCCT)

VCCT [36,37], which is based on the principle of linear elastic fracture mechanics (LEFM), is a well-known technique, like CZM, for simulating crack initiation and propagation along a pre-defined surface in 3D or a curve in 2D. VCCT is a method for the evaluation of SERR and mode-mixity ratio for cracks and assumes that the strain energy to be required for a new crack-face extension is equivalent to the one required for closing the same crack back to its original length in a 2D FE model in either plane-stress or plane-strain conditions; for example, Figure 2 shows the node separation by using VCCT when simulating the pure mode fracture. An initial crack front splits into two nodes *l*_1_ and *l*_2_ to form a new crack front at node *i*, while *u* and *v* denote the displacements in the direction of *x* and *y*, respectively. The SERR for pure mode I and II fracture based on VCCT are computed as follows:(7)$$G_{I}=\frac{1}{2t\delta_{a}}F_{y}(v-v^{\prime })\;G_{II}=\frac{1}{2t\delta_{a}}F_{x}(u-u^{\prime })$$
where *t* and $\delta_{a}$ denote the width and length of a crack-tip element, respectively. $F_{x}$ and $F_{y}$ are the shear and normal forces acting on the node *i*. In order to separate the node *i*, the following relation under mixed-mode loading condition must be satisfied:(8)$$f=\frac{G_{eq}}{G_{eqC}}\geq 1$$
where $G_{eq}$ is the equivalent energy-release rate computed at the node and $G_{eqC}$ is the critical equivalent energy-release rate computed based on different mixed-mode criterions such as Benzeggagh–Keane (B-K) law and power law which are used in this work.

## 3. Numerical Model

### 3.1. FE Model of Composite Single Lap Joint

In this paper, the commercial finite element software Abaqus [41] is used for predicting the load-displacement curves of SLJs. Although a 3D FE model could obtain more accurate results, a 2D model is generally preferred for parametric studies. To validate the proposed studies, the FE model of SLJ was built and compared to the experimental data from the literature [35]. The geometry and dimensions of the SLJ are shown in Figure 3. A lay-up of [0]_16_ layers of unidirectional carbon fiber-reinforced polymer (CFRP) composite laminates with a thickness *t_p_* of 2.4 mm and a length *L_p_* of 100 mm was utilized as an adherend. The material properties of the CFRP material are listed in Table 1. The SLJ model consists of two composite adherends, each with a trial specimen width of 25 mm, bonded together in the middle using two different types of adhesives: brittle adhesive Araldite^®^ AV138 and ductile adhesive Araldite^®^ 2015. The mechanical properties of the two types of adhesives are presented in Table 2. The bonding arrangement and boundary conditions are depicted in Figure 3, where *L_o_* represents the adhesive bonding length of 25 mm, and *t* is the adhesive thickness of 0.1 mm. The left side of the model is pinned with a length of 25 mm in the *x* and *y* directions, while the right side of the model is roller with a length of 25 mm in the *y* direction to only move in the axial direction for tensile loading on displacement control.

A four-node plane-strain continuum element with reduced integration (CPE4R) is used for the adherend mesh. The mesh of adhesive layer is built by using either the cohesive element (COH2D4) or the CPE4R, depending on different modeling strategies, which will be discussed later in Section 4. In the present work, the mesh size of 0.15 mm in the present SLJ model is selected as a standard mesh size that was already verified in the literature [35]. The influence of the mesh size on result accuracy for different modeling strategies must be considered. Therefore, the coarser, standard, and finer mesh sizes (0.5 mm, 0.15 mm, 0.05 mm) are chosen in the following analysis and performed in Section 4, in which different built-in damage approaches in Abaqus are considered for failure-load predictions of SLJ, i.e., cohesive element with zero/nonzero thickness, surface-based cohesive behavior, and VCCT. A detailed description of each method in Abaqus is presented in the following subsections.

### 3.2. Cohesive Element in Abaqus

The commercial FE software Abaqus has a built-in CZM capability, which has been widely used in the simulation of composite bonded joints and different-material interface failure due to many advantages such as easy implementation in conventional FE framework and without need of pre-crack.

The material properties of adhesives needed for prediction of damage initiation and propagation by using cohesive element are shown in Table 1, which is assigned to the section of the adhesive layer. In Abaqus, the cohesive element can be built as zero thickness or nonzero thickness depending on the practical applications. When the adhesive layer is modelled by using the cohesive element, a single layer of four-node 2D cohesive elements (COH2D4) is created and a tie constraint is applied at the interface between the adhesive layer and adherents. In cohesive elements, normal- and tangential-stiffness components are interpreted as penalty stiffness *K*, which is modified considering the thickness of the adhesive layer in the defined traction–separation section as well as in the elastic and shear modulus. The options of stress-based failure criterions for CZM in Abaqus have the maximum nominal stress (MAXS) and the quadratic nominal stress (QUADS) while the strain-based failure criterions are the maximum nominal strain (MAXE) and quadratic nominal strain (QUADE). The QUADS criterion is used in this article. Although different shapes of CZM laws (triangular, linear-exponential, and trapezoidal) are available in Abaqus, the triangular law shape is often good enough to describe crack growth as shown in Figure 1. To define the evolution of damage under mixed-mode behavior in terms of the fracture energy after the damage initiation, Abaqus provides the power law and B-K law that govern failure under mixed-mode conditions. The brief description of the cohesive element setting in Abaqus is given, and the reader is referred to the Abaqus documentation [41] for more specific details.

### 3.3. Surface-Based Cohesive Behavior in Abaqus

The surface-based cohesive behavior is another built-in cohesive approach in Abaqus. In this approach, the interaction between the adhesive layer and the adherents is modelled by cohesive contact instead of using elements, and its constitutive damage law is the same, with the cohesive element approach based on the traction–separation law. The constitutive damage response between the adherent and the adhesive is assigned as a surface-to-surface contact in Abaqus setting. The brief description of the cohesive contact setting in Abaqus is given, and the reader is referred to the Abaqus documentation [41] for more specific details.

Although both cohesive elements described in Section 3.2 and cohesive contact behavior in this section are very similar and valuable approaches for modeling damage interactions between surfaces in Abaqus, the selection between them depends on the required level of accuracy, the complexity of material behavior, and the computational resource available. Cohesive elements are more suitable for accurately capturing detailed interface behavior, while cohesive contact offers a more efficient approach when contact interactions are of primary interest and cohesive effects are secondary.

### 3.4. VCCT in Abaqus

VCCT is based on Irwin’s [42] crack-closure integral and is utilized when plastic dissipation does not exist. This approach can be easily implemented in the finite element framework, and the relevant energy-release rates can be computed by using the VCCT method employed in Abaqus. In the setup of VCCT crack in Abaqus, the VCCT-fracture criterion needs to be defined in the contact interaction property, and the surface-to-surface contact interaction is then created to model the potential crack surfaces using master and slave contact surfaces. The initially bonded region defines a region of the slave surface that is initially bonded with the master surface. The unbonded portion of the slave surface behaves as an initial crack using a regular contact surface. As explained earlier, an initial crack is needed when the VCCT is employed for modeling crack propagation or delamination. The present SLJ model is, however, pristine and has no initial crack. To accommodate the requirement of the VCCT, one element length of initial crack is created in the model. The theory of VCCT is described in Section 2.2, and more details of VCCT setup is referred to the Abaqus documentation [41].

## 4. Result and Discussion

### 4.1. Modeling Strategies for Nonzero Thickness of Adhesive

In the context of the present SLJ model assumed with an actual adhesive thickness of 0.1 mm, four different damage-modeling strategies are performed to predict the failure load of the SLJ model. The first modeling strategy is to model the adhesive layer by using the cohesive element of 0.1 thickness directly as shown in Figure 4a. The other three modeling strategies are to model the adhesive layer by using the continuum elements (CPE4R), where two interfaces between the adhesive and adherend are connected by using the cohesive element of zero thickness, surface-based cohesive behavior, and VCCT as shown in Figure 4b–d, respectively. Two different types of adhesive materials (brittle adhesive Araldite^®^ AV138 and ductile adhesive Araldite^®^ 2015) are considered in the model to see the influence of failure-load prediction using these modeling strategies.

Figure 5a demonstrates that the first modeling strategy, in which the adhesive layer is modeled by using the cohesive element of 0.1 thickness, accurately predicts the failure load for the adhesive 2015. The grey color area in the figures indicates the reference [35] with experimental results of the ductile adhesive 2015 ranging from about 9150 N to 10,750 N, which is applied to the following related figures in the article. The numerical results obtained from three different mesh sizes all fall within the experimental range. The predictions of other two modeling strategies corresponding to Figure 4b,c show limited mesh sensitivity as depicted in Figure 5b,c. It is shown that, for adhesive 2015, the failure-load predictions obtained with different mesh sizes conform to the experimental values and are well within the range of experiments. However, for the VCCT-modeling strategy corresponding to Figure 4d, none of the failure-load predictions with different mesh sizes in Figure 5d are located within the experimental range. A remarkable discrepancy between the failure-load prediction and the average experimental value can be observed in this case.

Figure 6 shows the failure-load predictions for the adhesive AV138. It can be seen that Figure 6a–c, corresponding to the modeling strategies of cohesive law in Figure 4a–c, underscore the substantial sensitivity to different mesh sizes for AV138 material. The grey color area in the figures indicates the reference [35] experimental results of the brittle adhesive AV138 ranging from about 6500 N to 7500 N, which is applied to the following related figures in the article. Noted that all the predictions do not fall within the experimental range except for the case in Figure 6a with a mesh size of 0.05 mm. However, Figure 6d highlights an interesting phenomenon observed with the VCCT-modeling strategy corresponding to the Figure 4d. In contrast to adhesive 2015, the failure-load predictions for adhesive AV138 consistently align with the experimental range in this VCCT case.

### 4.2. Modeling Strategies for Zero Thickness of Adhesive

In much research, the thickness of adhesive is considered in the FE model for detailed modeling of SLJ. However, the thickness of adhesive in the present SLJ model [35] is 0.1 mm, which is very thin compared to the thickness of the adherend. We want to know the effect if the thickness of adhesive is ignored in the SJL model when the thickness of adhesive is comparative thin. Therefore, the adhesive thickness is assumed to be zero in this subsection. Three different damage-modeling strategies are performed to predict the failure load of the SLJ model, including surface-based cohesive behavior, a zero-thickness cohesive element, and the VCCT corresponding to the Figure 7a–c, respectively. Brittle adhesive Araldite^®^ AV138 and ductile adhesive Araldite^®^ 2015 are considered in the model.

As shown in Figure 8a,b for adhesive 2015, it can be observed that both the surface-based cohesive-contact behavior and the zero-thickness cohesive element exhibit relatively low sensitivity to mesh size, and the failure-load predictions obtained from these two methods are within the experimental range. However, as indicated in Figure 8c, the VCCT demonstrates an inability to predict the failure load regardless of the chosen mesh size.

Turning to the adhesive AV138, it is evident that the surface-based cohesive behavior in Figure 9a and the zero-thickness cohesive element in Figure 9b result in an overestimation of the failure load for a mesh size of 0.5 mm. The other two finer mesh sizes, however, do not significantly affect the failure-load prediction. Finally, Figure 9c illustrates that the overall results of the VCCT approach very slightly exceed the experimental values, but it also shows a better mesh-sensitivity result compared to the other two modeling strategies as well as a more accurate prediction than the one for adhesive 2015 in Figure 8c.

### 4.3. Comparisons

In this section, the comprehensive findings from Section 4.1 and Section 4.2 are integrated and presented through a focused investigation for the comparison of brittle and ductile adhesives between seven different damage-modeling strategies (Figure 4 and Figure 7). To delve deeper into the influence of the seven different modeling approaches, each corresponds to a fixed mesh size of 0.15 mm in observing the load-displacement curve for both Adhesive 2015 and Adhesive AV138.

In Figure 10a, it can be observed that among the seven modeling strategies for ductile adhesive 2015, the failure-load predictions for both VCCT approaches with zero and nonzero adhesive thickness significantly exceed the range of experimental values. In addition, sudden-drop failure curves are observed that do not capture the force-displacement curve of ductile adhesive obtained from the experiments [35]. On the other hand, although the other five CZM modeling strategies exhibit subtle discrepancies regarding the failure-load prediction regardless of the adhesive thickness, they are all in good agreement with the failure load and the force-displacement curve in experiments [35]. Thus, it can be observed that, regardless of whether the adhesive thickness is considered, the VCCT approach appears to be less suitable than the CZM approach for predicting the failure load of the ductile adhesive. It is suggested that these five CZM modeling strategies all predict the failure load of ductile adhesive without observing the significant effect of mesh sensitivity.

Similarly, the investigation is extended to the case of brittle adhesive AV138 where the seven modeling strategies for predicting failure load and force-displacement response are evaluated as shown in Figure 10b. When the adhesive thickness is considered, the results of two CZM modeling strategies significantly exceed the range of experimental value and provide conservative predictions, whereas the case of 0.1 mm cohesive element has good predictive ability but significant mesh sensitivity that must be considered in the analysis, as seen in Figure 6a. The VCCT approaches gives more accurate predictions without the issue of mesh sensitivity. When zero adhesive thickness is considered, on the other hand, the results of three modeling strategies show that all have good predictive ability. Overall, it can be seen that, regardless of whether brittle adhesive thickness is considered, the usage of CZM modeling strategies is able to provide accurate predictions but needs to carefully consider the effect of mesh sensitivity. However, the VCCT approaches appear to have lower mesh sensitivity than the CZM approaches and can also provide good failure-load prediction when the brittle adhesive is considered.

## 5. Conclusions

In this work, progressive failure analyses in composite single-lap joints with a thin adhesive layer subjected to tensile loading are modeled by different built-in damage-modeling techniques in Abaqus, including CZM and VCCT. The influences of zero/non-zero adhesive thickness and brittle/ductile adhesives on failure-load prediction are investigated by using seven different damage-modeling strategies.

In the case of ductile adhesive 2015, the VCCT simulated results show that a remarkable discrepancy between the failure-load prediction and the experimental value can be observed regardless of whether the adhesive thickness is considered. The VCCT methodology appears to be less suitable than the CZM methodology for predicting the failure load. However, all the CZM modeling strategies are shown to be in good agreement with the experimental result for the failure-load prediction without observing the significant effect of mesh sensitivity.

For the case of brittle adhesive AV138, it can be seen that, regardless of whether the brittle adhesive thickness is considered, all the CZM modeling strategies are able to provide precise predictions but there needs to be careful consideration of the effect of mesh sensitivity. In contrast to the observation regarding ductile adhesive, the VCCT modeling strategies, however, have the capability to provide a good failure-load prediction and appear to have lower mesh sensitivity than the CZM modeling strategies.

The findings in the present work provide a deeper understanding of the merits and drawbacks of each modeling strategy. In future work, the influence of a thick and thin adhesive layer on failure-load prediction using different modeling strategies should be planned for investigation in order to further guide the design of adhesive bonded joints.

## Figures and Tables

**Figure 1 polymers-16-00964-f001:**
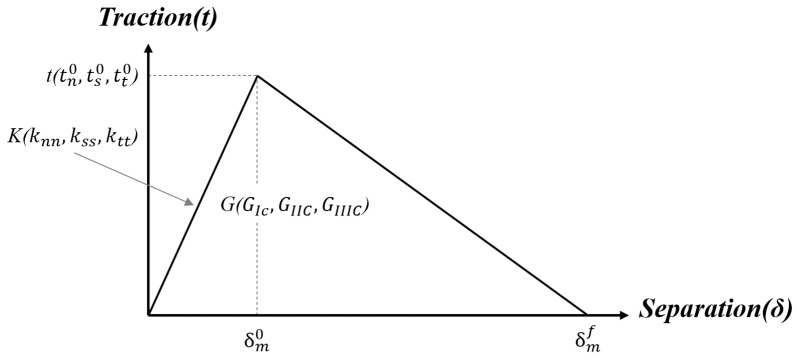
Bilinear cohesive zone model.

**Figure 2 polymers-16-00964-f002:**
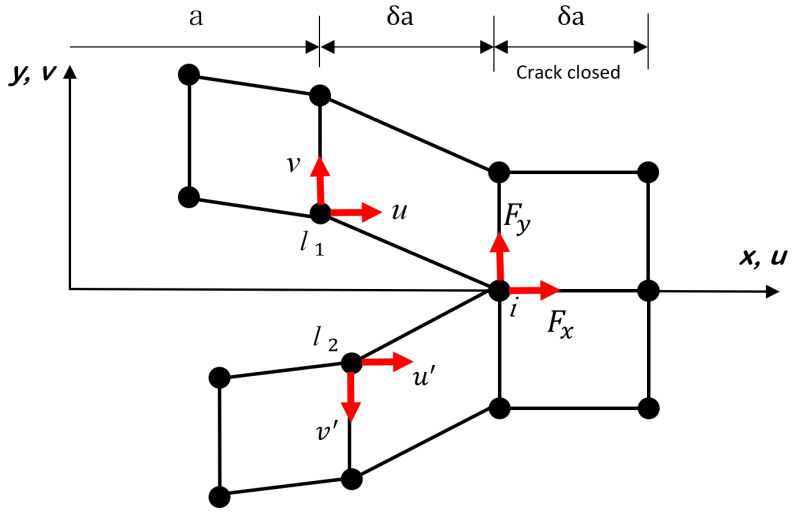
Schematic of virtual crack closure technique.

**Figure 3 polymers-16-00964-f003:**
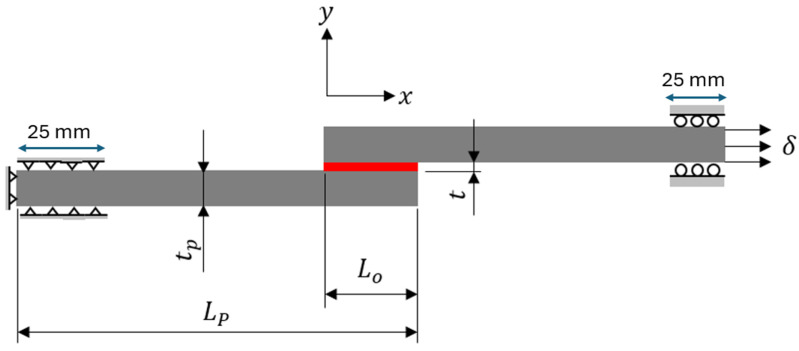
Geometries and dimensions of single-lap joints.

**Figure 4 polymers-16-00964-f004:**
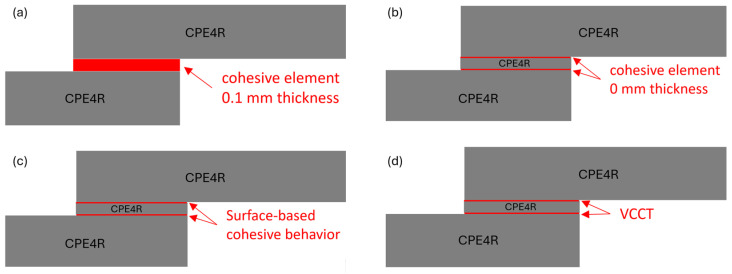
Modeling strategies for nonzero adhesive thickness. (**a**) Cohesive element with 0.1 mm thickness. (**b**) Cohesive element with 0 mm thickness. (**c**) Surface-based cohesive behavior. (**d**) VCCT.

**Figure 5 polymers-16-00964-f005:**
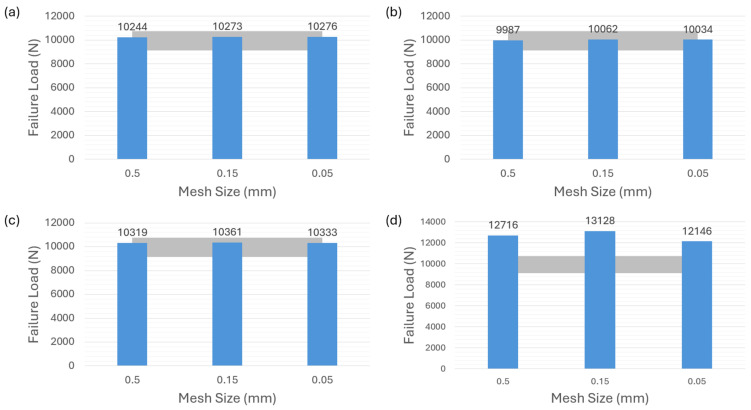
Failure-load predictions of ductile adhesive 2015 with nonzero thickness for different modeling strategies. (**a**) Cohesive element with 0.1 mm thickness. (**b**) Cohesive element with 0 mm thickness. (**c**) Surface-based cohesive behavior. (**d**) VCCT.

**Figure 6 polymers-16-00964-f006:**
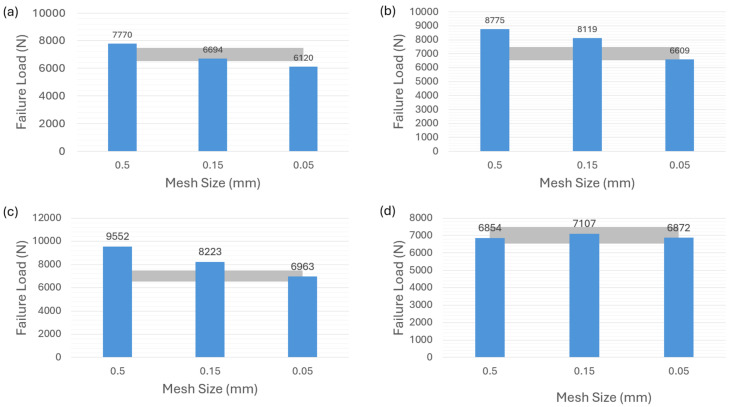
Failure-load predictions of brittle adhesive AV138 with nonzero thickness for different modeling strategies. (**a**) Cohesive element with 0.1 mm thickness. (**b**) Cohesive element with 0 mm thickness. (**c**) Surface-based cohesive behavior. (**d**) VCCT.

**Figure 7 polymers-16-00964-f007:**
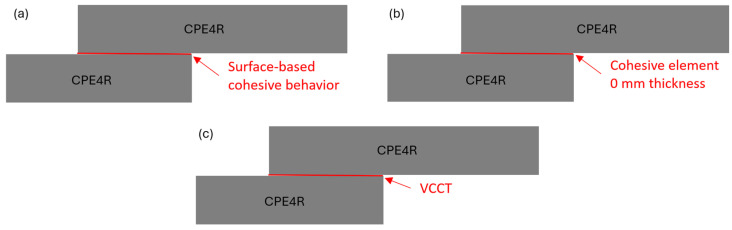
Modeling strategies for zero adhesive thickness. (**a**) Surface-based cohesive behavior. (**b**) Cohesive element with 0 mm thickness. (**c**) VCCT.

**Figure 8 polymers-16-00964-f008:**
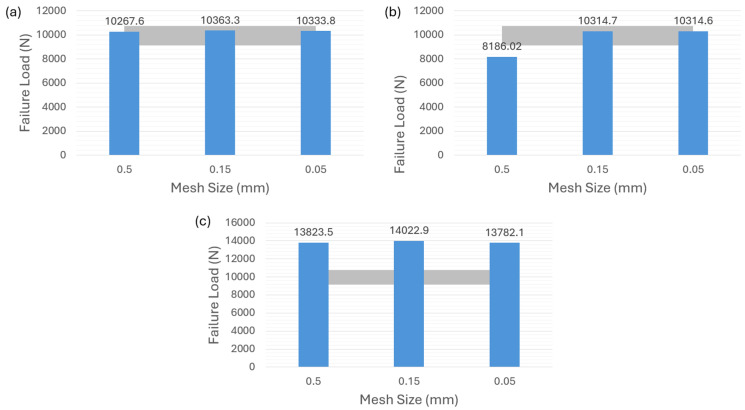
Failure-load predictions of ductile adhesive 2015 with zero thickness for different modeling strategies. (**a**) Surface-based cohesive behavior. (**b**) Cohesive element with 0 mm thickness. (**c**) VCCT.

**Figure 9 polymers-16-00964-f009:**
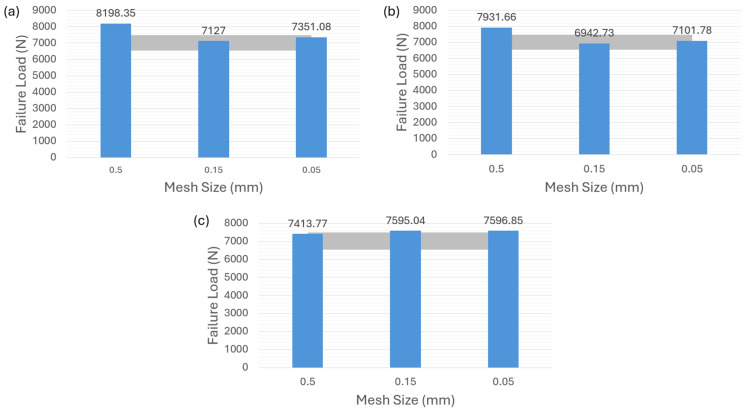
Failure-load predictions of brittle adhesive AV138 with zero thickness for different modeling strategies. (**a**) Surface-based cohesive behavior. (**b**) Cohesive element with 0 mm thickness. (**c**) VCCT.

**Figure 10 polymers-16-00964-f010:**
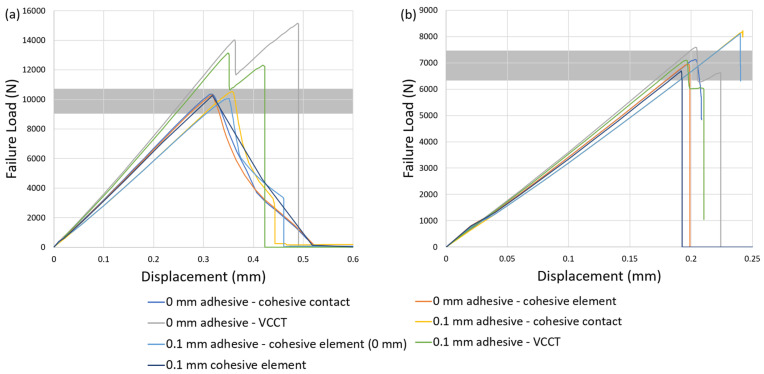
Failure-load comparisons for (**a**) ductile and (**b**) brittle adhesives between seven damage-modeling strategies.

**Table 1 polymers-16-00964-t001:** Material properties of CFRP.

*E*_11_ (MPa)	*E*_22_*, E*_33_ (MPa)	*G*_12_*, G*_13_ (MPa)	*G*_23_ (MPa)	*v* _12_ *, v* _13_	*v* _23_
109,000	8819	4315	3200	0.342	0.38

**Table 2 polymers-16-00964-t002:** Material properties for two types of adhesives.

Properties	Araldite^®^ AV138	Araldite^®^ 2015
E (GPa)	4.89	1.85
G (GPa)	1.56	0.56
$t_{n}^{0}\;(MPa)$	39.45	21.63
$t_{s}^{0}\;(MPa)$	30.2	17.9
$G_{n}^{0}\;(N/mm)$	0.2	0.43
$G_{s}^{0}\;(N/mm)$	0.38	4.7

## Data Availability

Data are contained within the article.

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
