# Peer review of "Influence of Failure-Load Prediction in Composite Single-Lap Joints with Brittle and Ductile Adhesives Using Different Progressive-Damage Techniques"

_polymers, 2024, doi:10.3390/polym16070964_

Round 1
Reviewer 1 Report
Comments and Suggestions for Authors
Overall, the article provides a comprehensive exploration of different damage modeling techniques for predicting the failure behavior of composite single-lap joints subjected to tensile loading. The inclusion of both CZM and VCCT methodologies, as well as the consideration of zero and non-zero adhesive thicknesses and brittle/ductile adhesives, adds depth to the analysis.
Overall, the article provides a comprehensive exploration of different damage modeling techniques for predicting the failure behavior of composite single-lap joints subjected to tensile loading. The inclusion of both CZM and VCCT methodologies, as well as the consideration of zero and non-zero adhesive thicknesses and brittle/ductile adhesives, adds depth to the analysis.
Here are some comments and suggestions for the authors:
1. The article effectively elucidates the CZM and VCCT methodologies, encompassing their underlying principles as well as their execution within Abaqus. Nevertheless, expounding upon the precise parameters and configurations employed in the simulations would be advantageous in order to assist readers who may lack familiarity with these methodologies.
2. The evaluation of outcomes across various modeling strategies is enlightening and yields significant knowledge regarding the merits and drawbacks of each methodology. An expanded discourse regarding the rationales behind the superior or inferior performance of particular modeling strategies in particular scenarios would be advantageous. Such an analysis would aid readers in comprehending the fundamental elements that impacted the outcomes.
3. To ensure that numerical simulations are reliable, an evaluation of mesh sensitivity is vital. The article alludes to mesh sensitivity in passing in a few sections; however, it would be advantageous to include a more comprehensive discourse on the influence of mesh size on result accuracy and the methods employed by the authors to mitigate or address potential complications associated with mesh sensitivity.
4. Incorporating a discourse on the pragmatic ramifications of the discoveries would augment the significance of the study. In what ways might the knowledge acquired from this research contribute to the development and evaluation of composite frameworks utilized in practical engineering contexts, with a specific focus on the aerospace sector?
5. Contributing to the article by proposing potential directions for future research grounded in the present findings would be beneficial. Have any unresolved inquiries or areas requiring development been identified in this research that may be explored in subsequent inquiries?
By providing a comprehensive examination of damage modeling methods for composite joints, the article contributes significantly to the body of knowledge in this area. By making a few minor improvements and providing additional clarifications, the article could be enhanced even more in terms of its impact and readability.
Comments on the Quality of English LanguageThere are occasional grammatical errors and punctuation issues scattered throughout the text. A careful proofreading to correct these errors would enhance the overall quality of the writing.
Author Response
The authors would like to thank the reviewer for the valuable comments and suggestions. Please see the attachment for the author's response.

Reviewer 2 Report
Comments and Suggestions for Authors
For this type of work, it would be better to have a comparison between the experimental results to the analytical modeling. I have few comments for the authors. Please consider these in revising the manuscript.
· It is recommended to add the applications of such lap joints with brittle and ductile adhesives in the introduction.
· In page 2, line 93, a total of six references are cited which is not good. It is expected to discuss briefly the outcomes from those researches rather than just referring.
· Briefly discuss the support condition used in Figure 3. What will happen if different support rather than fixed support is used?
· In abstract it is mentioned that the modelling data was compared with the experimental data reported in the literature. However, there is no table of figures in this regard. Please add a comparison table or figure to clarify this issue.
· How was the mesh size chosen? Please discuss briefly.
· Please rewrite the conclusion part by highlighting the major findings in bullet points.
Author Response

(The authors gave the same response as above.)

Round 2
Reviewer 2 Report
Comments and Suggestions for Authors
The authors have improved the manuscript. I have no more comments.